

# Application of the flipped classroom model based on Bloom's Taxonomy of Educational Objectives in endodontics education for undergraduate dental students

Yaru Wei[1] and Zhengjun Peng[2]

[1] Guanghua School of Stomatology, Guanghua School of Stomatology, Hospital of Stomatology, Sun Yat-Sen University, Guangdong Province Key Laboratory of Stomatology, Guangzhou, China

[2] Department of Operative Dentistry and Endodontics, Guanghua School of Stomatology, Hospital of Stomatology, Sun Yat-Sen University, Guangdong Province Key Laboratory of Stomatology, Guangzhou, China

Corresponding author
Zhengjun Peng,
pengzhj3@mail.sysu.edu.cn

## ABSTRACT

**Introduction:** The current study was conducted to compare the effects of the lecture method of teaching and the flipped classroom model based on Bloom's Taxonomy of Educational Objectives on the teaching of endodontics curriculum to undergraduate students majoring in stomatology, and to develop a standardized teaching process based on the flipped classroom model.

**Materials and Methods:** A standardized flipped classroom model based on Bloom's Taxonomy of Educational Objectives was established. Two groups of undergraduate students majoring in stomatology received instruction in a portion of the endodontics curriculum using either the lecture method or flipped classroom model of teaching. A teaching questionnaire was administered to evaluate the students' mastery of theoretical knowledge, understanding of learning objectives, satisfaction of teaching method, and learning interest. The SPSS 26.0 software was used for statistical analysis, and the $t$-test was used to compare the differences between the two groups.

**Results:** Both learning model cohorts filled out assessment questionnaires upon completion of the pilot curriculum. Compared with the responses from students in the lecture-based group, the self-rating of theoretical knowledge reported by students in the flipped classroom cohort increased by 10.9%, from $7.1 \pm 0.8$ to $7.9 \pm 0.7$ ($t = 2.912$, $p < 0.006$). Students' test scores in the flipped classroom group increased by 17.1%, from $7.0 \pm 0.8$ to $8.2 \pm 0.7$ ($t = 4.284$, $p < 0.001$). Students' understanding of ideological and humanistic objectives as well as medical ethics were both significantly improved by 11.4% ($t = 2.267$, $p = 0.009$) and 13.9% ($t = 2.600$, $p = 0.014$), respectively. Students' satisfaction with the teaching model and class duration increased significantly, by 11.1% ($t = 2.782$, $p = 0.009$) and 14.3% ($t = 2.449$, $p < 0.020$), respectively. Students' learning interest increased by 17.1% ($t = 3.101$, $p = 0.004$). The length of study time prior to class under the flipped classroom model was longer than when using the traditional lecture method ($t = 3.165$, $p = 0.003$), but the flipped classroom model shortened review time after class ($t = 4.038$, $p = 0.001$).
Students' self-reported understanding of teaching objectives improved by 8.3% ($t = 1.762$, $p = 0.083$), and satisfaction with the preview method and curriculum increased by 8.1% ($t = 1.804$, $p = 0.081$) and 11.1% ($t = 1.861$, $p = 0.072$), respectively. There was no statistically significant difference between the two groups.

**Conclusions:** The flipped classroom teaching model based on Bloom's Taxonomy of Educational Objectives, combined with humanistic teaching objectives, can improve the efficacy of instruction, and merits popularizing and applying in the teaching of undergraduate students majoring in stomatology.

# INTRODUCTION

In the 1950s, Ralph W. Tyler stated that educational objectives are the foundation and core of the curriculum, and that the curriculum should be based on the establishment of educational objectives (*Tyler, 2008*). Bloom's Taxonomy of Educational Objectives, proposed by educator B.S. Bloom's team in 1956, divides educational objectives into three main areas: cognitive, affective and skill-oriented (*Bloom, Krathwohl & Masia, 1964*). At the beginning of the 20th century, *Anderson et al. (2001)* described the objectives of the cognitive domain and established the six dimensions of the cognitive process: reading, comprehension, application, analysis, evaluation and creativity. Traditional curriculum divides teaching objectives into three categories: mastery, familiarity and comprehension, but these objectives may not reflect those of the humanities. According to Bloom's Taxonomy of Educational Objectives, the classification of the educational objectives of a course can clarify teaching goals and ideas, rationalize teaching content, make humanistic content more natural to students, improve student empathy, and establish high quality medical ethics, while strengthening the effect of theoretical teaching and effectively combining theory-based and humanistic education.

Lectures represents a traditional form of teaching. When theoretical content of a curriculum is extensive and classroom time is limited, it can be a challenge for teachers to efficiently, clearly, and engagingly convey theoretical knowledge. The flipped classroom model can effectively extend learning time. Using this model, students independently watch an explanatory video based on the curriculum in advance, allowing for classroom time to be used more efficiently for teacher-student and student-student interaction, answering questions and solving problems, and for collaboration and exploration, ultimately leading to improved teaching and learning outcomes (*Cui et al., 2023*; *Lage, Platt & Treglia, 2000*; *Li, Tang & Cheng, 2023*; *Moquin et al., 2023*).

Traditionally, theoretical instruction for students majoring in stomatology is organized around a lecture-based curriculum. Students taught under this model reported that the learning objectives were broad and vague, the theoretical knowledge was difficult to

understand and the teaching style lacked interest and engagement. Therefore, an exploration of innovative teaching methods is crucial for the optimization of undergraduate theoretical teaching.

*Ordu, Aydoğan & Çalışkan (2024)* conducted a study to determine the effect of an interactive learning method using prepared questions based on Bloom's Taxonomy on nursing students' learning of the need for movement, and the authors recommended that studies on interactive learning be repeated in different subjects within nursing education. Another study found that Generative Pre-trained Transformer 4 (GPT-4) demonstrated a strong performance, in regards to Bloom's Taxonomy, when queried with psychosomatic medicine multiple-choice exam questions (*Herrmann-Werner et al., 2024*). An investigation into teaching reform, led to the proposal of an innovative flipped classroom model based on Bloom's Taxonomy of Educational Objectives to be applied in the theoretical teaching of endodontics. The present study aimed to compare the flipped classroom model with the curriculum-based lecture method by utilizing questionnaires to compare and analyze students' learning levels in terms of theoretical knowledge and learning outcomes under the two teaching models. An additional goal was to develop a standardized flipped classroom model that is applicable for undergraduate stomatology curriculum, aiming to implement it in undergraduate education across the country.

## MATERIALS AND METHODS

### Study participants and design

Study participants were divided into two groups, each consisting of sixteen third-year students enrolled in the 5-year undergraduate stomatology program at Guanghua School of Stomatology, Sun Yat-sen University. The study was approved by the Ethics Committee of the Hospital of Stomatology, Sun Yat-sen University (No. KQEC-2024-72-01). This approval qualifies for a waiver of review and does not require the signing of an informed consent form. The educational material used for the study was the nineteenth chapter of the national textbook of higher education, "Endodontics," which covers root canal therapy (5th edition, People's Health Publishing House). The study duration was 1 year.

### Teaching methodology

Sixteen students received theoretical lessons employing a curriculum-based lecture method (Model 1). Under this model students reviewed the curriculum and textbook material prior to class, while the instructor delivered theoretical concepts in the classroom through lectures, including the use of visual presentation aides. The instructor summarized key points at the end of the class and engaged students with questions and answers during and after the lecture. The presentations and audio were recorded in real-time and uploaded to an online platform so that students could review and reflect on the curriculum after class.

A second group of sixteen students received theoretical lessons using a flipped classroom model (Model 2) based on Bloom's Taxonomy of Educational Objectives. The

lecturer for both groups was the same professor from the endodontics program who possessed 5 years of dentistry teaching experience. The Endodontics Teaching and Research Department at Guanghua School of Stomatology, Sun Yat-sen University, categorized the teaching objectives of the pilot curriculum into three main areas: knowledge, competence and emotion, with the knowledge area focusing on learning and elaboration of theoretical knowledge, the competence area focusing on clinical application of knowledge, and the emotion area focusing on humanistic goals (Table 1). The content of Chapter 19 of "Endodontics" was divided into five theoretical modules, for which the Department of Endodontics Education and Research produced five corresponding micro-videos: (1) Overview of the development of root canal therapy and case selection, (2) Anatomical morphology of the medullary cavity, (3) Root canal preparation and sterilization, (4) Root canal obturation and (5) Strategies for the prevention and treatment of complications in root canal therapy. Students reviewed the curriculum based on Bloom's Taxonomy of Learning Objectives and watched the micro-video before class. During this time, students had the opportunity to independently work on the learning objectives in the knowledge domain, as well as reflect on the learning objectives in the skills and affective domains, ensuring alignment with classroom instruction. In the classroom, the instructor imparted the basic concepts and enhanced the content of the knowledge areas. This was followed by group discussions where the teacher posed questions about the curriculum content. Students responded in groups, and the teacher reviewed the responses for omissions, provided additional information and summarized key points. This process helped clarify the teaching and learning objectives. The teacher concluded by summarizing the discussion using presentations and board notes. The overall focus in the classroom was to provide instruction and facilitate interactive discussions about the objectives related to skill and affective areas. Classroom lessons were also recorded in real-time and uploaded to an online platform for students to review after class. At the end of the lesson, students had the opportunity to review and reflect on the content in relation to the curriculum based on Bloom's Taxonomy of Educational Objectives.

## Questionnaire survey

After completing the pilot curriculum, both groups of students completed a questionnaire involving examining, comparing and analyzing the degree of mastery of the theoretical content of this chapter, understanding of the learning objectives, satisfaction with the teaching methods, learning interest and duration and additional metrics (Fig. 1). Feedback on the degree of satisfaction and acceptance of the two teaching modes was collected using a five-point Likert scale as follows: very dissatisfied/disagree (1); dissatisfied/disagree (2); unsure (3); satisfied/agree (4); very satisfied/agree (5). Feedback on students' knowledge acquisition was measured using a 10-point scale, with 10 representing a perfect score and 6 indicating a sufficient score. To compare learning outcomes, the study calculated students' final exam grades for the chapter exam questions and converted the grades of the two groups to a 10-point scale to standardize analysis.

**Table 1 Learning objectives for endodontics.**

| Classification of learning objectives | Curriculum guidelines |
|---|---|
| Knowledge objectives | 1. Concepts of root canal therapy |
| | 2. Overview of the development of root canal therapy |
| | 3. Principles of root canal therapy and case selection |
| | 4. Root canal therapy procedure |
| | 5. Strategies for the prevention and treatment of common complications in root canal therapy |
| Competency objectives | 1. Describe the challenges that the anatomical complexity of the root canal system poses to the cleaning and disinfection of the root canal? |
| | 2. Describe the methods that must be taken to achieve the most complete elimination of infection? |
| | 3. What do doctors and patients need to do to prevent a microleak from developing after root canal therapy that leads to failure? |
| | 4. Describe the analysis of the causes and treatment of pain that occurs during or after root canal therapy? |
| | 5. Describe the causes and analysis of complications that occur during root canal therapy? |
| Emotional goals | 1. How do I communicate with patients who need root canal therapy and how do I present the program? |
| | 2. Why is it important for endodontists to learn about root canal therapy? |

## A normalized teaching process for the flipped classroom model based on Bloom's Taxonomy of Educational Objectives

The flipped classroom model based on Bloom's Taxonomy of Educational Objectives for the theoretical teaching of "Endodontics" Chapter 19, Root Canal Therapy, in the 5-year undergraduate dental program was developed by the Department of Endodontics, Guanghua School of Stomatology, Sun Yat-sen University. After repeated discussions and modifications, a standardized teaching process was developed (Fig. 2). This process is summarized as follows:

1. Instead of the traditional lecture-based curriculum, the teaching objectives were divided into three main parts: knowledge, skills, and emotion. These were provided to students for previewing 1 week before class.

2. Teaching content was optimized by dividing the theoretical knowledge of the chapters into five modules. A corresponding micro-video was created for each module and uploaded to an online platform 1 week before the course, and the teaching secretary ensured that all students completed the pre-course study.

3. Students were divided into four groups to discuss and learn from the questions in modules 1, 2, 4, and 5. Module 3 was an extension not discussed in class.

4. In the classroom, the instructor first taught basic concepts and other theoretical content through lecture. Fragmented content, basic definitions, *etc.*, did not need to be presented in the form of micro-videos. During the teaching modules, questions were posed to students to encourage critical thinking and response. The teacher then provided corrections and summarized any mistakes made by the students.

5. The students asked questions, which were answerd by the teacher before the end of the lesson, and then repeated and summarized through presentations.

Questionnaire on the effectiveness of teaching with the flipped classroom
method in the bachelor's degree program in dentistry

Department of Endodontics, Guanghua School of Stomatology, Sun Yat-sen
University, China

Dear Classmates:

First of all, thank you for completing this questionnaire. The results are crucial for us to improve our teaching methods. On behalf of the faculty, we would like to thank you for your cooperation. In the field of Endodontics, specifically in root canal therapy, which is characterized by a multitude of complex concepts and challenges in comprehension, we are actively seeking and exploring innovative teaching methods to simplify the course material, enhance interactively, and make the learning experience more enjoyable. We have created this questionnaire to investigate the effectiveness of teaching. Please read the questionnaire carefully and try to select answers that are realistic.

Name:__________Genders:__________Grade:__________Classes:__________

1. At the end of this chapter, you have evaluated the content of this chapter on root canal therapy as follows (10 out of 10, 6 passed).

| 10 | 9 | 8 | 7 | 6 | 5 | 4 | 3 | 2 | 1 |
|----|----|----|----|----|----|----|----|----|----|

2. At the end of this chapter, your knowledge of the topic "overview of the development of root canal therapy and case selection" will be assessed as follows (10 out of 10, 6 passed).

| 10 | 9 | 8 | 7 | 6 | 5 | 4 | 3 | 2 | 1 |
|----|----|----|----|----|----|----|----|----|----|

3. At the end of this chapter, your knowledge of the anatomy of the medullary cavity is assessed as follows (10 out of 10, 6 passed).

| 10 | 9 | 8 | 7 | 6 | 5 | 4 | 3 | 2 | 1 |
|----|----|----|----|----|----|----|----|----|----|

4. At the end of this chapter, your knowledge of "root canal preparation and sterilization" will be assessed as follows (10 out of 10, 6 passed).

| 10 | 9 | 8 | 7 | 6 | 5 | 4 | 3 | 2 | 1 |
|----|----|----|----|----|----|----|----|----|----|

5. How would you rate your knowledge of "root canal filling" in this chapter? (10 out of 10, 6 passed).

| 10 | 9 | 8 | 7 | 6 | 5 | 4 | 3 | 2 | 1 |
|----|----|----|----|----|----|----|----|----|----|

6. At the end of this chapter, your knowledge of "strategies for prevention and treatment of complications in root canal therapy" will be assessed as follows (10 out of 10, 6 passed).

| 10 | 9 | 8 | 7 | 6 | 5 | 4 | 3 | 2 | 1 |
|----|----|----|----|----|----|----|----|----|----|

7. The syllabus will help you to understand the aims and key points of the chapter.

    a) □Very agree    b) □Agree    c) □Unsure    d) □Disagree    e) □Very disagree

8. The syllabus will help you understand the ethical, and humanistic teaching objectives of the chapter.

    a) □Very agree    b) □Agree    c) □Unsure    d) □Disagree    e) □Very disagree

9. Learning and mastering the contents of this chapter will facilitate your growth as a doctor with medical ethics, competence and cordiality.

    a) □Very agree    b) □Agree    c) □Unsure    d) □Disagree    e) □Very disagree

10. This course will awaken your interest in Endodontics.

    a) □Very agree    b) □Agree    c) □Unsure    d) □Disagree    e) □Very disagree

11. Are you satisfied with the teaching mode in this chapter?

    a) □Very agree    b) □Agree    c) □Unsure    d) □Disagree    e) □Very disagree

12. Were you satisfied with the way you reviewed this chapter before class?

    a) □Very agree    b) □Agree    c) □Unsure    d) □Disagree    e) □Very disagree

13. Are you satisfied with this part of the curriculum?

    a) □Very agree    b) □Agree    c) □Unsure    d) □Disagree    e) □Very disagree

14. Are you satisfied with the duration of the course (2 credit hours)

    a) □Very agree    b) □Agree    c) □Unsure    d) □Disagree    e) □Very disagree

15. The total time you spent studying before class was_______minutes.

16. The total time you spent reviewing after school was_______minutes.

17. If you have any further comments or suggestions on the teaching chapter, please fill in this form

____________________________

Thank you very much for your cooperation!

**Figure 1** **Questionnaire on the effectiveness of teaching with the flipped classroom method.**

6. Questionnaires examining the effectiveness of the lessons were collected 1 week after the course.

7. Teaching was optimized based on student feedback and expert evaluations.

## Statistical analysis

SPSS 26.0 software was used to conduct statistical analysis on the questionnaire results from the two groups of students. Comparison of the learning outcomes of the pilot chapter with the learning outcomes of modules 1–4, the results of the theoretical examination, understanding of learning objectives and key points, comprehension of humanities objectives, grasp of medical ethics and morals, interest in learning endodontics, satisfaction with the teaching method, pre-study methods, curriculum, teaching hours and a comparison of study hours prior to class were conducted using chi-square variances and

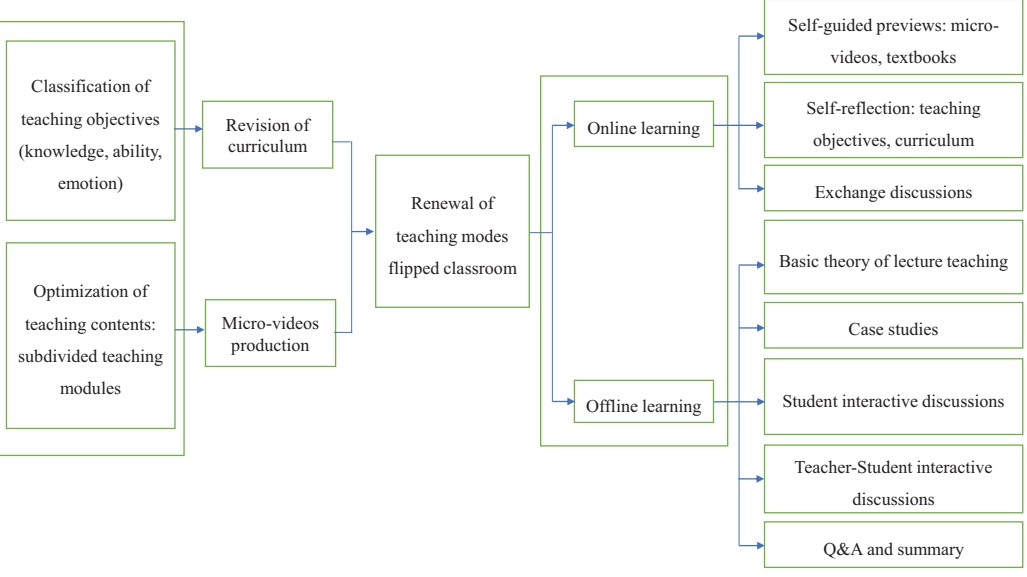

**Figure 2 Establishment of a standardized teaching process.**

t-tests for independent samples between two groups of data. Comparison of module 5 learning outcomes and post-course learning hours was performed with t-tests for independent samples to assess heterogeneity of variance between the two groups of data. Analyses were evaluated at a significance level of $\alpha = 0.05$.

## RESULTS

### Comparison of the effectiveness of the two teaching models

A total of sixteen students from the lecture method cohort (Model 1) attended the course, and all 16 questionnaires were subsequently completed, representing a response rate of 100%. The students' personalized comments and suggestions on teaching included: "Less class time, more content, lectures are too fast," "I hope animations and anatomical modeling are integrated," "The course content is difficult," "I hope that more clinical aspects are included," "I suggest adding more lessons," "The theory is abstract, animations could enhance understanding" and "I recommend incorporating pictures or 3D models for better comprehension".

A total of sixteen students from the flipped classroom cohort (Model 2) participated in the course, and all 16 questionnaires were subsequently completed, resulting in a response rate of 100%. Students' personalized comments and suggestions about the class included: "I hope the curriculum is more detailed," "Increase the number of class hours, otherwise the content density is too high," "Pictures can be added" and "I hope the introduction of new concepts can be more concrete and vivid".

Statistical analysis of the student questionnaire revealed that students in the flipped classroom cohort showed a 10.9% increase in theoretical knowledge learning compared to students in the lecture-based cohort ($7.9 \pm 0.7$ *vs*. $7.1 \pm 0.8$; Table 2). This difference was statistically significant ($p < 0.006$). The scores of the five theory modules in the flipped

**Table 2 Comparison of feedback results from questionnaires in two teaching modes.**

| Iterms | Mode 1 | Mode 2 | t-value | p-value |
|---|---|---|---|---|
| 1. Study score of chapter root canal therapy (points) | 7.1 ± 0.8 | 7.9 ± 0.7 | 2.912 | 0.006[b] |
| 2. Module 1 "Overview of the development of root canal therapy and case selection" learning score (points) | 7.1 ± 0.8 | 7.8 ± 0.5 | 3.177 | 0.003[b] |
| 3. Module 2 "the anatomy of the medullary cavity" learning score (points) | 6.9 ± 0.8 | 7.8 ± 0.5 | 3.707 | 0.001[b] |
| 4. Module 3 "root canal preparation and sterilization" learning score (points) | 6.8 ± 0.8 | 7.7 ± 0.7 | 3.696 | 0.001[b] |
| 5. Module 4 "root canal filling" learning score (points) | 7.0 ± 0.7 | 7.7 ± 0.5 | 2.825 | 0.008[b] |
| 6. Module 5 "strategies for prevention and treatment of complications in root canal therapy" learning score (points) | 6.9 ± 0.9 | 7.6 ± 0.5 | 2.777[a] | 0.009[b] |
| 7. Theoretical examination results (points) | 7.0 ± 0.8 | 8.2 ± 0.7 | 4.284 | 0.000[b] |
| 8. Understanding of learning objectives and key points (points) | 3.6 ± 0.6 | 3.9 ± 0.5 | 1.762 | 0.083 |
| 9. Understanding of humanistic goals (points) | 3.5 ± 0.5 | 3.9 ± 0.6 | 2.267 | 0.038[b] |
| 10. Understanding of medical ethics (points) | 3.6 ± 0.5 | 4.1 ± 0.6 | 2.600 | 0.014[b] |
| 11. Interest in studying endodontics (points) | 3.5 ± 0.5 | 4.1 ± 0.6 | 3.101 | 0.004[b] |
| 12. Satisfaction with teaching mode (points) | 3.6 ± 0.5 | 4.0 ± 0.4 | 2.782 | 0.009[b] |
| 13. Satisfaction with the way in which the pre-test was conducted (points) | 3.7 ± 0.6 | 4.0 ± 0.6 | 1.804 | 0.081 |
| 14. Satisfaction with the curriculum (points) | 3.6 ± 0.5 | 4.0 ± 0.6 | 1.861 | 0.072 |
| 15. Satisfaction with the duration of the course (points) | 3.5 ± 0.5 | 4.0 ± 0.6 | 2.449 | 0.020[b] |
| 16. Length of pre-course preparation (min) | 32.5 ± 9.30 | 42.8 ± 9.12 | 3.165 | 0.003[b] |
| 17. Length of after-school revision (min) | 84.4 ± 31.1 | 50.0 ± 13.8 | 4.038[a] | 0.001[b] |

Note:
Mode 1 is a lecture-teaching model based on the curriculum, and mode 2 is a flipped classroom model based on Bloom's taxonomy of educational objectives. A ten-point scale was used for items 1–7 and a five-point scale was used for items 8–15. The label a implies that the variance between the two groups is not homogeneous, using the t' test. [b]$p < 0.05$ indicates a statistically significant difference between the two groups.

classroom cohort were higher by 9.8%, 13.0%, 13.2%, 10.0% and 10.1%, respectively, with an average increase of 11.2%, representing a statistically significant difference ($p < 0.01$). In the final theory exam, students' exam scores in the flipped classroom group increased from 7.0 ± 0.8 to 8.2 ± 0.7, with an average improvement of 17.1% ($p < 0.001$). At the same time, students' understanding and appreciation of humanistic goals and medical ethics increased significantly, by 11.4% ($p = 0.038$) and 13.8% ($p = 0.014$), respectively. Students' satisfaction with the mode of instruction and the duration of teaching also increased significantly, by 11.1% ($p = 0.009$) and 14.3% ($p < 0.020$), respectively. Students' interest in learning increased significantly, by 17.1% ($p = 0.004$). Pre-class study time was approximately 10 min longer for the flipped classroom group compared to the lecture group ($p = 0.003$). However, post-class study time in the flipped classroom group was reduced by approximately 30 min compared to the lecture group ($p = 0.001$), a statistically significant difference. An analysis was conducted of the post-class study time after pre-class study as the same time (minutes) in mode 1 (35.5 ± 9.16) and mode 2 (36.14 ± 9.23). The results showed that the flipped classroom model (54.6 ± 11.7) was able to reduce post-class study time compared to the lecture method (85.5 ± 18.9), a statistically significant difference ($p < 0.01$) (Data not shown). In the flipped classroom group, students' understanding and appreciation of the teaching goals increased by 8.3% compared to the lecture group ($p = 0.083$), and students' satisfaction with the preparation

method and curriculum increased by 8.1% ($p = 0.081$) and 11.1% ($p = 0.072$), respectively, compared to the lecture group. There was no statistical difference among the three groups mentioned above.

## DISCUSSION

Theoretical teaching is characterized by abundant textual knowledge, often presented in a manner that some students may consider abstract and unengaging. The instructor may incorporate pictures and animations to aid student learning and understanding, but due to classroom time constraints, it can be difficult to strike a satisfactory balance between providing sufficiently detailed explanation and remaining on the academic schedule. Flipped classroom instruction using micro-videos, combined with the characteristics of case-based learning, can effectively extend the classroom before and after the scheduled class time (*Li et al., 2023*). Micro-videos are brief and concise, employing targeted division of theoretical knowledge, with each video addressing one topic with strong focus, facilitating learning and understanding of the teaching objectives. Micro-videos present theoretical knowledge in a vivid and fresh manner in order to stimulate and attract interest in learning (*Kaushik et al., 2023*). Micro-video length is generally 10 min or less, tailored to students' attention span, as well as their physical and mental development characteristics. The availability of micro-videos online enables students to self pace, aiding independent study and review. The core concept of the flipped classroom, as a novel education reform model, is to remove course content from the classroom and allow students to explore and learn independently outside the classroom (*Uchida et al., 2022*). In the present study, the flipped classroom model divides and records the learning content into micro-videos for independent pre-class viewing. Maximum student initiative is encouraged while valuable classroom time focuses on addressing questions and points of confusion. Students engage more actively in project-based learning, collaboratively researching and solving problems, which can promote the internalization of knowledge (*Fatima et al., 2019*). Improved learning efficiency in endodontics undergraduate teaching have yielded positive results in promoting self-directed learning among medical students, consistent with findings from other related professional practices (*French et al., 2020*; *Hew & Lo, 2018*).

Since 2017, the university's endodontincs program has strove to establish a close connection between knowledge-and-skills-based teaching and values-based education into the curriculum. How to efficiently integrate the content of humanities education into undergraduate teaching, how to organicallly combine theoretical knowledge, clinical practice, and humanities education and how to promote innovations in teaching represent challenges in undergraduate education when considering the target curriculum content and requirements established by professors. Along with the professional curriculum of endodontics, additional focus should be placed on strengthening education on medical ethics and professional demeanor. Emphasis should be placed on education regarding medical benevolence, ensuring that students comprehend the importance of prioritizing the maintenance of oral and maxillofacial health, function and aesthetics, while continuously honing their medical skills. Moreover, it is crucial to prioritize respecting patients during clinical diagnosis and treatment to enhance these students' ability to

empathize. Continuous improvement of doctor-patient communication skills is crucial in building professional trust and reliablity.

Bloom's Taxonomy of Educational Objectives divides teaching objectives into three major categories and six dimensions. By incorporating the teaching of theoretical knowledge and the construction of humanistic education in the curriculum, it effectively provides guidance for teachers to clarify teaching objectives, teaching ideas and the design of academic content. Sanghee Yeo analyzed the response to integrated courses of a medical school that introduced an outcome-based curriculum, and found that most of the verbs used to describe course outcome belonged to the two lower levels of Bloom's taxonomy: knowledge and comprehension (*Yeo, 2019*). Pedro Tadao Hamamoto Filho et al. investigated the psychometric properties of items according to their classification in Bloom's taxonomy and judges' estimates, employing an adaptation of the Angoff method, and they found that items with high-level taxonomy performed better in discrimination indices, and additionally that a panel of experts may develop coherent reasoning regarding the difficulty of items (*Hamamoto Filho et al., 2020*). These studies confirmed that Bloom's taxonomy can be applied effectively to humanities curriculum.

Aiming to address the pain points and challenges present in current theoretical teaching, the Department of Endodontics at Guanghua School of Stomatology, Sun Yat-sen University, has implemented teaching reforms. This includes the introduction of a flipped classroom teaching model based on Bloom's Taxonomy of Educational Objectives to help standardize the teaching process and methodology. Using this model, syllabus objectives are categorized into targeted micro-videos that are easily digestible for students. This approach enables students to engage in independent learning before class. Using this system, students can understand the knowledge, skill and emotional objectives of the curriculum prior to class. In the classroom, students are guided to discuss and study the key concepts, helping them to generalize and summarize, with mistakes corrected promptly by the teacher. The purpose of this model is to encourage students to be more receptive to the theoretical content and to stimulate their interest in learning through discussion and interaction. In the classroom, learning objectives are conveyed through clinical cases, doctor-patient communication and other practical scenarios. After class, students can reinforce the teaching objectives by reviewing the syllabus within the context of Bloom's Taxonomy of Educational Objectives.

The current study employed a questionnaire to assess the efficacy of the two teaching models. The students' self-assessment of theoretical knowledge improved, which was reflected in improved exam scores. Additionally, students' self-reported understanding of humanistic objectives, medical ethics and morals significantly increased. Moreover, students' satisfaction with the teaching methods and class duration also improved. Analysis of the questionnaire responses found that the flipped classroom model improved students' understanding of the teaching objectives, although the improvement was not statistically significant. This result was mirrored in the satisfaction survey of the pre-study process and the syllabus. The methodology of this study presented certain limitations. Due to the academic setting, it was not feasible to randomly divide students of the same grade into two groups, and there existed a memory bias. However, analysis of the questionnaire

and the results of the final examination can still objectively and realistically reflect the efficacy of the teaching models.

Teaching reform aims to enhance teaching effectiveness without increasing the burden on students and teachers. As technology advances, increasingly sophisticated tools are being integrated into traditional teaching methods, including virtual simulations (*Meng et al., 2023*), micro-videos (*Li, Tang & Cheng, 2023*), apps (*Deng et al., 2023*) and artificial intelligence tools such as ChatGPT (*Herrmann-Werner et al., 2024*). The present study found that, under the flipped classroom model, the total study duration, both pre- and post-class, did not differ significantly, but the post-class duration was significantly shorter compared to the pre-class duration. A potential explanation for this is that students spent sufficient time before class to understand, organize and reflect on the teaching objectives. Due to students' understanding and mastery of the course objectives, as well as adequate communication with the teacher and fellow classmates in the classroom, less time was required for post-class review and study.

For teachers, it is essential to invest more time and effort in fostering teaching innovation and implementing teaching reform. This could include creating short and concise micro-videos before classes, as well as revamping lesson plans and syllabi. Teachers can update their teaching concepts and innovate their teaching methods in line with teaching reform. The department can compile a series of micro-videos through this reform, which serve as a valuable teaching resource for online and offline instruction, catering to undergraduate students, graduate students, and international students.

Syllabus-based lectures represent a classic teaching method with certain strengths and weaknesses. Students find pre-course preparation relatively easy, while teachers can effectively control the classroom flow, rhythm and time. Additionally, course materials and syllabi are often repeated during multiple iterations of the same course, which reduces preparation time. However, this teaching method often leads to average performance in terms of student participation, teacher-student interaction, classroom engagement and teaching effectiveness. In comparison, the teaching approach of the flipped classroom model based on Bloom's Taxonomy of Educational Objectives offers several advantages. These include clear teaching goals, improved integration of humanities teaching, innovative and engaging teaching methods, enhanced student interaction and participation, dynamic and engaging classroom environments and positive teaching outcomes. However, students are required to engage in more pre-class thinking and discussion, as well as increase motivation and active participation in classroom discussions and interactions. Additionally, this model requires teachers to invest more time and effort in class preparation and face challenges in controlling the pace and timing of classroom discussions and interactions. Though the flipped classroom model requires teachers to adapt their skillset, this new teaching approach could improve their effectiveness and help them better manage their classroom teaching time.

It is necessary to acknowledge this work's limitations. The small sample size of the study limits the impact and scope of the results. Therefore, subsequent studies should increase the sample size to verify the observed findings. Future studies could also incorporate

teaching aides, such as AV (audio/video) and tooth simulation molds to enrich the classroom teaching content.

## CONCLUSION

The current study has resulted in the development of a standardized flipped classroom teaching model based on Bloom's Taxonomy of Educational Objectives. This model, combined with humanistic teaching objectives, can increase the efficacy of teaching, and is worth popularizing and applying in the theoretical teaching of undergraduate students majoring in stomatology.

### Funding

This research was funded by Teaching Quality and Teaching Reform Project of Sun Yat-sen University (2023) and Teaching Quality and Teaching Reform Project for Undergraduate Colleges and Universities in Guangdong Province (2023). The funders had no role in study design, data collection and analysis, decision to publish, or preparation of the manuscript.

### Grant Disclosures

The following grant information was disclosed by the authors:
Teaching Quality and Teaching Reform Project of Sun Yat-sen University (2023).
Teaching Quality and Teaching Reform Project for Undergraduate Colleges and Universities in Guangdong Province (2023).

### Competing Interests

The authors declare that they have no competing interests.

### Author Contributions

- Yaru Wei conceived and designed the experiments, performed the experiments, analyzed the data, prepared figures and/or tables, and approved the final draft.
- Zhengjun Peng conceived and designed the experiments, performed the experiments, authored or reviewed drafts of the article, and approved the final draft.

### Human Ethics

The following information was supplied relating to ethical approvals (*i.e.*, approving body and any reference numbers):

Medical ethics committee of hospital of stomatology sun yat-sen university approval to carry out the study within its facilities (Ethical Application Ref: KQEC-2024-72-01).

### Data Availability

The raw measurements are available in the Supplemental File.

## Supplemental Information

Supplemental information for this article can be found online at http://dx.doi.org/10.7717/peerj.18843#supplemental-information.

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
