# Peer review of "Application of the flipped classroom model based on Bloom’s Taxonomy of Educational Objectives in endodontics education for undergraduate dental students"

_PeerJ, doi:10.7717/peerj.18843_

## Round 0.1 · original submission · Minor Revisions

Thank you for an interesting article, and apologies for the long delay in processing. It has been challenging to find sufficient reviewers.

I agree that minor revisions are required. In particular, please consider the limitations of the study and consider citing more studies.

·

Basic reporting

This manuscript has clear hypotheses and solid results.

Experimental design

The test and study design are excellent. The only question is whether the survey is in English or another language. If it is in another language, please provide the original copy of the survey sample.

Validity of the findings

The findings are clear, and the statistics are very impressive.

Additional comments

This is a very good article to demonstrate flipped classroom mode. Minor adjustment is suggested.

Reviewer 2 ·

Basic reporting

no comment

Experimental design

no comment

Validity of the findings

no comment

Additional comments

1. How about the limitation of the study.
2. Pre-class study time was about 10 minutes longer in mode 2 than in mode 1(p = 0.003). However, mode 2 was able to reduce post-class study time by about 30 minutes (p = 0.001), a statistically significant difference. Relying on questionnaires to calculate study time is not very accurate.

Reviewer 3 ·

Basic reporting

It is a very thorough and detailed research paper. The majority of the article has a clear English language use, please review the document for grammatical errors, e.g. -
1. line 94 'Ethics Committee of Hospital of stomatology, Sun Yat-sen university' to match the text written in line 93
2. line 108, 'received' to receive
3. Please consider using terms like 'Obturation' instead of root canal filling while explaining steps for Root Canal Therapy/Endodontic Therapy.
4. Scoring method in lines 139-142 can be clarified better; 'dissatisfied/agree' can be confusing; rather use the complete terms 'dissatisfied/disagree'
5. Line 216 - 'This research' to The current research in education

Experimental design

No comment

Validity of the findings

No comment

Reviewer 4 ·

Basic reporting

Dear Authors,
The manuscript. Titled "Application of flipped classroom mode based on Bloom's Taxonomy of Educational Objectives in endodontics teaching of undergraduate dental students ". The current manuscript, in general, needs several corrections; please find below the comments to improve the manuscript.

Abstract:
1. Kindly provide MeSH keywords so the article becomes more accessible to search after publication.

Introduction:
1. The introduction starts with the problem statement; kindly start with the literature instead of beginning with the problem statement.
2. The introduction is not up to the mark. Kindly elaborate more and add more studies.

Experimental design

Methods:
1. Kindly provide the sample size calculation.
2. Mention the duration of the study.

Validity of the findings

Results:
1. In the result section, the methodology is written as objective. It should be written only in the materials and method section.

Additional comments

Discussion:
1. Kindly add more studies to the discussion.
2. Kindly provide the limitations of the study
3. Kindly add future study ideas
Conclusion:
Should be elaborated and written as per the objectives

References:
1. Kindly add more recent references.

---

## Round 0.2 · Minor Revisions

Thank you for the revisions you have made so far, the manuscript is much improved.

In addition to the edits made so far for content, before Acceptance, I would suggest that you consider editing for English to improve clarity for the reader. You may consider doing this independently. Alternatively, PeerJ can provide language editing services if you wish - please contact them at [email protected] for pricing (be sure to provide your manuscript number and title). Your revision deadline is always extended while you undergo language editing.

---

## Round 0.3 · accepted · Accept

Thank you for your work in amending this article for publication, I'm now happy to accept it for publication.